# Immunological Pathomechanisms of Spongiotic Dermatitis in Skin Lesions of Atopic Dermatitis

**DOI:** 10.3390/ijms23126682

**Published:** 2022-06-15

**Authors:** Ryoji Tanei, Yasuko Hasegawa

**Affiliations:** 1Department of Dermatology, Tokyo Metropolitan Geriatric Hospital and Institute of Gerontology, 35-2 Sakaecho, Tokyo 173-0015, Japan; 2Department of Geriatric Pathology, Tokyo Metropolitan Geriatric Hospital and Institute of Gerontology, 35-2 Sakaecho, Tokyo 173-0015, Japan; yhasegaw@tmig.or.jp

**Keywords:** apoptosis, atopy patch test, cell-mediated cytotoxicity, eczematous dermatitis, Fas/Fas-ligand system, IgE-mediated delayed-type hypersensitivity, inflammatory dendritic epidermal cells, Langerhans cells, lichenified eczema, spongiosis

## Abstract

Atopic dermatitis (AD) is a chronic pruritic skin disease with a complex pathogenesis underlying its heterogeneous clinical phenotypes and endotypes. The skin manifestation of AD reflects the cytokine milieu of a type-2-dominant immunity axis induced by genetic predisposition, innate immunity dysregulation, epidermal barrier defects, and allergic inflammation. However, the detailed pathomechanism of eczematous dermatitis, which is the principal characteristic of AD, remains unclear. This review examines previous studies demonstrating research progress in this area and considers the immunological pathomechanism of “spongiotic dermatitis”, which is the histopathological hallmark of eczematous dermatitis. Studies in this field have revealed the importance of IgE-mediated delayed-type hypersensitivity, the Fas/Fas-ligand system, and cell-mediated cytotoxicity in inducing the apoptosis of keratinocytes in spongiotic dermatitis. Recent studies have demonstrated that, together with infiltrating CD4 T cells, IgE-expressing dendritic cells (i.e., inflammatory dendritic epidermal cells and Langerhans cells) that capture specific allergens (i.e., house dust mites) are present in the spongiotic epidermis of lichenified eczema in patients with IgE-allergic AD. These findings suggest that IgE-mediated delayed-type hypersensitivity plays a pivotal role in the pathogenesis of spongiotic dermatitis in the skin lesions of AD.

## 1. Introduction

Atopic dermatitis (AD), a chronic, relapsing eczematous dermatitis that occurs frequently in infants and children, is characterized by a complex pathogenesis and heterogeneous clinical phenotypes [1,2]. AD is a pruritic skin disease that develops and progresses in association with a variety of factors, such as dry skin and epidermal barrier dysfunction due to genetic abnormalities (e.g., filaggrin gene mutations) or non-hereditary chronic inflammation [2,3,4,5], innate immunity dysregulation [6,7], dominance of T helper (Th) 2 cytokine (i.e., interleukin [IL]-4 and IL-13) immunity [2,8,9,10], and intractable and diverse itching [11,12,13]. The characteristics of AD (e.g., phenotypes and endotypes, clinical course, and disease severity) are associated with several factors, including patient age [14,15,16], race [15,17], genetic predisposition [18], skin dysbiosis involving increased predominance of *Staphylococcus aureus* (*S. aureus*) [19,20], sex hormones [21], and environmental stimuli (e.g., allergens, irritants, and chemicals) [22]. Recent research has revealed that prolongation, recurrence, or new onset of AD occurs not only in adults [23,24], but also in the elderly [25,26,27]; therefore, AD is considered to be an allergic disease that can become a lifelong condition. In addition, AD is now thought to involve cutaneous and systemic inflammation [28,29,30], and it is characterized as involving complications associated with both allergic diseases (i.e., asthma, allergic rhinitis, food allergies, etc.) and non-allergic diseases (hypertension, heart disease, diabetes, autoimmune disease, etc.) [29,31]. Furthermore, it has been suggested that persistent IgE sensitization, a central characteristic of AD, is related to the pathogenesis of hypertension [32] and carcinogenesis suppression [25,33,34].

This review first summarizes previous studies regarding the pathophysiology of AD, in which the above-mentioned complex factors could serve as the basis of the pathomechanism via the interactions between Th2-dominant cytokines and chemokines. We then review research related to the immunological pathomechanism of spongiotic dermatitis, which is the pathological hallmark of the eczematous dermatitis that characterizes AD, and propose a hypothesis regarding the underlying mechanism.

## 2. Pathophysiology of AD Based on the Type-2-Dominant Immunity Axis Cytokine Milieu

It has long been assumed that AD is a disease of Th2 immunity, as the main axis of the immunopathogenesis of AD involves Th2 cytokines (i.e., IL-4 and IL-13). This assumption was recently confirmed by demonstration of the anti-AD therapeutic efficacy of dupilumab, an anti–IL-4/IL-13 biologic drug targeting IL-4 receptor α (IL-4Rα), a common subunit of the IL-4 and IL-13 receptors [35,36]. The clinical phenotypes of AD are generally classified as either IgE-allergic AD or non–IgE-allergic AD, depending on the presence or absence of IgE allergy in the pathomechanism, but both phenotypes exhibit Th2-dominant immunity [37]. With regard to Th2 cytokines, IL-13, rather than IL-4, is the major cytokine secreted by Th2 cells in the skin lesions of AD [38,39,40,41]. However, the peripheral blood IL-4 concentration in AD patients [26] and the serum total IgE level (IL-4 primarily regulates its synthesis) in IgE-allergic AD patients [37,41] reportedly show a positive correlation with the severity of AD lesions. Therefore, IL-4 is now thought to regulate the central part of the Th2 immunity pathway in the pathogenesis of AD [41,42]. Consistent with the characteristics of AD, it was reported that serum levels of IL-4Rα in both IgE-allergic AD and non–IgE-allergic AD patients are significantly higher than IL-4Rα levels in healthy controls [43].

Recent studies have shown that Th1 cytokines (IL-2, interferon [IFN]-γ, etc.) and Th2 cytokines (IL-5, IL-13, etc.) involved in adaptive immunity are also produced by innate immune cells such as innate lymphoid cells (ILCs), including group 1 ILCs (NK cells and ILC1 cells) and group 2 ILCs (ILC2 cells). These cytokines are also collectively referred to as type-1 and type-2 cytokines [6]. In patients with AD, type-2 inflammation-promoting chemokines (including thymic stromal lymphopoietin [TSLP], IL-25, and IL-33) can be released from keratinocytes by the following pathogenesis: epidermal barrier dysfunction (due to genetic factors such as filaggrin mutations and extrinsic factors such as low humidity), environmental stimuli (e.g., scratching and dysbiosis of *S. aureus*), and recognition of the innate immunity activation signals via receptors (e.g., toll-like receptors). These type-2 inflammation-promoting chemokines activate innate effector cells that could release type-2 cytokines and chemokines, such as dendritic cells (DCs), mast cells (MCs), and ILC2 cells. In addition, TSLP can directly interact with Th2 cells (10). These data indicate that innate immunity plays an important role in the pathogenesis of type-2 immunity dysregulation and the linkage with additive immunity in AD skin lesions [6,7] (Figure 1).

In adaptive immunity, AD skin lesions are considered to be T cell-mediated biphasic immune reactions to exposure to environmental factors such as allergens (e.g., house dust mites [HDMs], pollen, and foods) or pathogens (e.g., *S. aureus*) (Figure 1). Immunohistochemical and molecular analyses have shown that immune activation of Th2 cells and Th2-derived cytokines (i.e., IL-4, IL-5, IL-13, and IL-31), Th22 cells and Th22-derived cytokines (i.e., IL-22), and Th17 cells and Th17-derived cytokines (i.e., IL-17) is involved in development of acute AD lesions, and progressive activation of Th2 and Th22 cells and their cytokines and significant upregulation of Th1 cells and Th1-derived cytokines (i.e., IFN-γ) are involved in chronic lesions [2,44,45,46,47]. Infiltration of Th1 cells into chronic lesions requires IL-12 secretion, and macrophages, eosinophils, and DCs are considered to be sources of this cytokine [45]. Selective recruitment of CC chemokine receptor (CCR)4-expressing Th2 cells is mediated by Th2 chemokines (e.g., CC chemokine ligand [CCL]17/thymus and activation-regulated cytokine [TARC], and macrophage-derived chemokine [CCL22]). Selective recruitment of CXC chemokine receptor (CXCR)3-expressing Th1 cells is mediated by Th1 chemokines (e.g., CXC chemokine ligand [CXCL]10). These chemokines are primarily secreted by macrophages (CCL17 and CCL22), DCs (CCL17 and CCL22), and keratinocytes (CCL22 and CXCL10) [42,47]. Th2 cytokines (i.e., IL-4 and IL-13) stimulate B cells to produce IgE and promote IgE-induced upregulation of the high-affinity IgE receptor (FcεRI) on MCs and DCs. The binding of IgE to receptors on DCs (e.g., Langerhans cells [LCs] and inflammatory dendritic epidermal cells [IDECs]) facilitates allergen uptake and triggers hypersensitivity reactions in IgE-allergic AD [1,20,47]. The Th2 cytokine IL-5 induces peripheral eosinophil activation. Th2 and Th22 cytokines (i.e., IL-4, IL-13, and IL-22) contribute to the non-genetic impaired expression of skin barrier proteins (e.g., filaggrin, loricrin, and involucrin) and barrier dysfunction in skin lesions. The Th22 cytokine IL-22 induces epidermal hyperplasia of chronic lesions in AD patients [2,48]. The Th2 cytokine IL-31 is associated with chronic itching, and IL-4 and TSLP also play a role in the induction of itching [12]. Compared to IgE-allergic AD, non–IgE-allergic AD exhibits greater activation of Th17 and Th22 cells and secretion of their cytokines [2,37]. In severe cases of IgE-allergic AD, progression to autoallergy to self-antigens in keratinocytes may also occur [47,49]. In addition, molecular immune activation in lesion-affected skin induces systemic inflammation and immune dysregulation, which, in turn, affects non-lesioned skin, especially in patients with moderate-to-severe AD. The expression profiles of cytokines and chemokines in non-lesioned skin are similar to those in lesion-affected skin in these patients [30,50].

The molecular pathomechanisms of AD are based on the complex linkage and network involving the above-mentioned cytokines and chemokines released by inflammatory infiltrating cells and keratinocytes [22]. However, the characteristics of AD as an allergic disease with eczematous dermatitis are not sufficiently described by pathomechanisms centering only on interactions between cytokines and chemokines. Therefore, this review also examines the immune pathomechanisms underlying AD from the perspective of allergic inflammation.

## 3. Allergic Inflammation in AD: Is IgE a Bystander?

IgE-mediated allergies to environmental allergens (including foods) have been incorporated into the most commonly used clinical AD criteria [51] as a diagnostic biomarker for AD. However, since the 1930s, when the concept of atopic (IgE) allergy involvement in AD was introduced [52], the role of IgE in the pathomechanism of AD has remained controversial [1,53]. Considerable clinical data suggest that IgE plays a significant role in the pathogenesis of AD [43]. Approximately 70–80% of adult AD patients show high serum total IgE and specific IgE levels, and the severity of skin lesions in IgE-allergic AD patients is correlated with serum total IgE levels [37]. In addition, clinical improvement of skin lesions in patients with IgE-allergic AD has been reported in a specific-IgE allergen-free environment [54]. However, a previous study proposed that, “a wheal induced by IgE-induced type I immediate allergy does not cause eczematous dermatitis of AD, which is a type IV allergic reaction”, which was particularly problematic upon the introduction of the concept of IgE allergy in AD [53]. Although this suspicion was almost resolved by the discovery of IgE-bearing LCs [55] and the introduction of the concept of IgE-mediated delayed-type hypersensitivity [56], the question still remains because IgE expression on LCs is not a feature specific to patients with AD; it also occurs in patients with other inflammatory skin diseases involving serum hyper-IgE [57]. The suspicion that “the involvement of specific IgE allergy in the pathogenesis of AD might be simply epiphenomenon” was driven by the observation that omalizumab, an anti-IgE biologic, is effective only in some patients, even if they are IgE-allergic AD patients [7,58]. On the other hand, dupilumab, an anti–IL-4/IL-13 biologic, is effective in most AD patients, regardless of IgE-allergic or non–IgE-allergic phenotype [15,33]. Therefore, at present, it is generally considered that the role of IgE allergy is not the major part in the AD pathogenesis, but type-2 inflammation (IL-4/IL-13, etc.) induced by genetic predisposition, innate immunity disturbance, and epidermal barrier defects is responsible for the central part of the AD pathogenesis [54,59]. However, several factors may underlie the inadequate efficacy of omalizumab. For example, in omalizumab nonresponders, blood levels of total IgE in patients with IgE-allergic AD may exceed levels that can be suppressed by the usual omalizumab dose, and IgE-independent pathomechanisms might be activated even in patients with IgE-allergic AD [60,61]. In addition, accumulated data regarding the effectiveness of anti-IgE treatments in recent trials (e.g., immunoadsorption of IgE [62,63,64], combined the use of omalizumab and immunoadsorption of IgE [65], and a double-blind, placebo-controlled randomized clinical trial of omalizumab in pediatric AD patients [66]) confirm the importance of IgE allergy in the pathogenesis of AD [1,60,61].

Importantly, it should be recognized that IgE-associated immune pathomechanisms are located downstream of multiple molecular pathomechanisms of AD, in which type-2 immunity (mediated by IL-4/IL-13) plays a central role. In addition, specific IgEs in IgE-mediated delayed-type hypersensitivity play an essential role in the development of eczematous dermatitis and function primarily as inducing and amplifying factors.

## 4. Pathomechanism of Eczematous Dermatitis in AD as Allergic Inflammation

### 4.1. IgE-Mediated Delayed-Type Hypersensitivity and Spongiotic Dermatitis in AD

In terms of the pathomechanism of eczematous dermatitis, AD can be regarded as a reaction similar to protein contact dermatitis or systemic-type contact dermatitis [67,68], although its immunopolarization involves a shift to Th2-type immunity (IL-4/IL-13, etc.). In addition, from the perspective of the pathomechanism of allergic inflammation, AD is characterized by a transition from standard (IgE-independent) delayed-type hypersensitivity (type IV allergy) in the early stage of the disease to complex forms of allergies involving an IgE-dependent immediate response (type I allergy) [48], IgE-mediated late-phase reaction [10,46,69], and IgE-mediated delayed-type hypersensitivity [10,55,56] in the advanced stage of the disease [1]. This transition of the immune pathomechanism of AD is similar to that of a mouse model of repeated application of a contact allergen, in which a transition to a type I reaction is observed when type IV allergy persists as chronic inflammation [70]. Recent immunohistochemical and molecular analyses have suggested that IgE-mediated and IgE-independent allergic inflammation may coexist in the pathomechanism of IgE-allergic AD [60]. Of these, the most characteristic immune pathomechanism of AD is IgE-mediated delayed-type hypersensitivity [56]. This reaction is a special form of type IV allergy exhibiting persistent chronic inflammation due to allergen stimulation that induces a shift from type IV allergy to a type I reaction and an increase in blood IgE production. This increase in IgE production induces the continuous expression of FcεR I on DCs (e.g., LCs) and surface expression of IgE; DCs take up allergens via IgE-FcεR I on the cell surface and present the allergen peptides (antigens) to T cells [55,56]. In this reaction, the response threshold to the allergens is lowered compared with standard delayed-type hypersensitivity; therefore, it becomes sensitive enough to activate an allergic immune response and possible presentation of an extremely large amount of allergens/antigens to T cells [56,71]. This means that allergen-specific IgE harbored by patients with IgE-allergic AD acts not only as an effector for type I and late-phase reactions (reactions that induce itching–scratching cycles by activating MCs and eosinophils, thus exacerbating AD) [47,69], but also as an inducer and amplifier of delayed-type hypersensitivity [56,71]. Hence, these IgE antibodies can be associated with the aggravation and intractability of AD [72]. Notably, unlike FcεR I expressed by MCs and basophils, FcεR I expressed by DCs lacks a classical β chain [56].

Delayed-type hypersensitivity in the skin causes eczematous dermatitis, leading to manifestations of AD, allergic contact dermatitis (ACD), and positive-reaction sites of atopy patch tests (APTs) or patch tests (PTs) performed to diagnose these diseases. The histopathological hallmark of eczematous dermatitis is “spongiotic dermatitis” exhibiting intercellular edema with mononuclear cell infiltration in the epidermis as “spongiosis” and inflammatory cell infiltration in the upper dermis as “dermatitis” [55,60,73,74]. Routine hematoxylin and eosin (H and E) staining and immunostaining do not show a significant difference in the tissue response of spongiotic dermatitis of AD versus other eczematous dermatitis conditions such as ACD [75]. However, it was recently demonstrated that allergen-specific IgE is involved in the pathomechanism of spongiotic dermatitis in patients with IgE-allergic AD, in which IgE-mediated delayed-type hypersensitivity is activated [76]. In our analysis of chronic active skin lesions of adult and elderly IgE-allergic AD patients with HDM allergy, IDECs expressing IgE gathered in the central area of the spongiosis lesion, accompanied by LCs and infiltrating lymphocytes. This was similar to the findings of spongiotic dermatitis observed in the positive APT reaction sites for HDM antigens, as the APT examines the induction of AD-like lesions on the non-lesioned skin of AD patients [77]. Furthermore, in our analysis, IgE+ IDECs present with HDM antigens were observed in the central area of the spongiotic epidermis of skin lesions in approximately 60% of examined cases; some of these IgE+ IDECs appeared to be in contact with CD4+ T lymphocytes [76]. These results, thus, strongly suggest that in IgE-allergic AD patients with HDM allergy, allergen-specific IgE-mediated delayed-type hypersensitivity is induced in the epidermis by intraepidermal invasion of HDM allergens, resulting in the development of spongiotic dermatitis. In other words, the data clearly show that similar to AD-like skin lesions caused by the APT for HDM allergy [77,78,79,80,81], skin lesions caused by exposure to HDMs actually occur in naturally appearing skin lesions of AD. The accumulation of DCs within sites of spongiosis was once thought to be due to LCs; thus, when they formed clusters, they were called “Langerhans cell microgranulomas” [82] or “Langerhans cell microvesicles” [83]. However, it is now understood that the accumulated cells are primarily IDECs, and they are therefore called “epidermal dendritic cell clusters” or “epidermal dendritic cell aggregates” [77,80]. Previously, the accumulation of IDECs in spongiosis was confirmed only in skin lesions of APT-positive reaction sites, which are considered prototype acute lesions of AD; however, this accumulation of IDECs in spongiosis lesions was also demonstrated in the active lesions of chronic AD. This suggests that the induction of acute tissue reactions (i.e., spongiosis formation) is not uncommon, even in chronic AD lesions [46].

### 4.2. Pathomechanism of IgE-Allergic AD Based on IgE-Mediated Delayed-Type Hypersensitivity as the Primary Axis

The following is a review of previous research on the pathophysiology of IgE-allergic AD in which HDMs serve as the primary allergen, as HDMs are the most representative environmental allergens associated with AD. Pathological conditions caused by HDMs may be similar to those caused by other aeroallergens, such as pollen.

#### 4.2.1. Pathophysiology of Skin Manifestations of IgE-Allergic AD

Although many details remain unclear, spongiotic dermatitis in AD, similar to other eczematous dermatitis conditions, such as ACD, keratinocyte apoptosis induced by accumulated DCs and T lymphocytes in response to allergens/antigens, is thought to play a crucial pathophysiologic role [84]. In IgE-allergic AD, IgE-mediated delayed-type hypersensitivity is thought to be prominently involved in the induction of apoptosis in this spongiotic dermatitis. The primary DC expression markers are well known: CD1a+ and CD207(langerin)+ for LCs; CD1a+, CD11b+, CD11c+, and CD206+ for IDECs, and CD11c+ for dermal inflammatory DCs [85]. In addition, some dermal inflammatory DCs exhibit the CD1a− CD11c+ and CD1a+ CD11c+ phenotypes, with the latter considered to correspond to IDECs that infiltrate the dermis [76,86]. This section is presented with a commentary on histopathological images (Figure 2a–d and Figure 3a–l).

##### Uninvolved Skin

The non-lesioned skin of IgE-allergic AD patients did not differ significantly from healthy normal skin of control subjects by routine H and E staining and appeared to be normal skin (Figure 2a). When observed by immunostaining, however, regular arrangements of LCs expressing IgE on the surface via FcεR I were observed in the epidermis (Figure 3a). These LCs significantly prolonged IgE-expressing dendrites, and some of them extended beyond the tight junctions to the area underneath the stratum corneum, as if ready to capture allergens invading from the skin surface. However, IgE expression on LCs is readily reduced by treatment with anti-inflammatory agents such as topical steroids, and a mixture of IgE+ and IgE− LCs is often observed in both non-lesioned and lesioned skin [86,87]. In addition, these IgE-expressing LCs in the epidermis are not specific to patients with IgE-allergic AD, but are also found in patients with other inflammatory skin diseases involving serum hyper-IgE [57,86]. Non-lesioned skin of IgE-allergic AD patients usually exhibits no IDECs in the epidermis and no or few infiltrating IgE+ inflammatory DCs in the dermis (Figure 3a) [86,88]. In addition, histopathological analysis of non-lesioned skin revealed almost no HDM antigens captured by IgE+ LCs in the epidermis and IgE+ infiltrating cells in the upper dermis (Figure 3b) [86,89]. However, compared with the healthy skin of control subjects, the numbers of IL-13-expressing T cells [44], IL-4-expressing T cells (CD4+ cells), INF-γ-expressing T cells (CD4+ cells and CD8+ cells), and DCs [79] among the infiltrating cells were significantly increased in the non-lesioned skin of IgE-allergic AD patients.

##### Non-Spongiotic Areas of Lichenified Eczema

The skin lesion that most closely reflects the characteristics of AD with a chronic and recurrent clinical course is lichenified eczema, which is listed as a diagnostic criterion [51]. In lichenified eczema of AD, it is presumed that non-specific innate immunity-induced inflammation caused by scratching and other environmental irritating factors is also involved in the pathomechanism [22,58]. However, the involvement of adaptive immunity in the pathomechanism of lichenified eczema of AD is clear. Compared with non-lesioned skin, among infiltrating cells in the skin lesions of lichenified eczema, the numbers of IL-13-expressing T cells [45], IL-4-expressing T cells (CD4+ cells), INF-γ-expressing T cells (CD4+ cells and CD8+ cells), and DCs [79] are significantly increased. Although lichenified eczema is generally considered to be a chronic AD lesion, histopathological findings can show both a non-spongiotic area (Figure 2b, non-spongiotic dermatitis) and a spongiotic area (Figure 2d, spongiotic dermatitis). However, the latter is present less frequently, and it is therefore often necessary to prepare serial sections to confirm the diagnosis [76]. In both areas, characteristics of chronic lesions are observed (e.g., epidermal hyperplasia with an irregular extension of the rete ridges, and dermal infiltration of lymphocytes and macrophages with increased MCs that can be mixed with eosinophils).

When the non-spongiotic area is evaluated by IgE-mediated immunopathology as the main histopathological area of lichenified eczema, the presence of both IgE+ LCs and IgE+ IDECs in the epidermis represents a characteristic finding (Figure 3c). However, there is a difference in localization between IgE+ LCs and IgE+ IDECs, as IgE+ LCs are often present in the middle to upper layers of the epidermis, whereas IgE+ IDECs are often present in the subepidermal layer (Figure 3c). In addition, the dendrites of each cell tend to extend in the vertical direction for IgE+ LCs and in the horizontal direction for IgE+ IDECs in the epidermis [90]. IgE expression on LCs tends to be lower than that observed in non-lesioned skin, and LCs that express minimal IgE are also mixed in the epidermis (Figure 3c). Regarding the detection of HDM antigens in the skin lesions of AD by histopathological analyses, it has been reported that approximately 60% of IgE-allergic AD patients with serum HDM-specific IgEs exhibit HDM antigens in the skin lesions at a significantly higher level than in patients with non–HDM-allergic AD [89]. In our clinical study, the number of HDM antigens observed in the epidermis was low in the non-spongiotic area of lichenified eczema, and most of these antigens were captured by IgE+ LCs (approximately five on average per 0.24 mm^2^ in a 200× visual field) (Figure 3d). In some cases, HDM antigens coexisting with IgE+ LCs were prominently observed in the perifollicular epidermis, which has been pointed out as an area of the stratum corneum with a weak barrier function [4]. In contrast, HDM antigens are also captured by IgE-expressing cells other than LCs in the papillary and upper layers of the dermis, such as dermal inflammatory DCs [76,86]. In the epidermis of non-spongiotic areas, IgE+ LCs that extend dendrites toward the tight junctions and capture HDM antigens are occasionally observed (Figure 3d) [76,86]. However, in this area, few lymphocytes satellite to IgE+ LCs with HDM antigens were observed in the epidermis; therefore, these HDM antigen-capturing IgE+ LCs are assumed to cause the skewing of Th2 inflammation via the production of cytokines and chemokines [56], but not to present HDM antigens to lymphocytes in the epidermis [76].

##### Spongiotic Dermatitis in APT Positive-Reaction Sites

By definition, acute lesions of AD are characterized by severe pruritic papules on extensively erythematous skin, accompanied by serous exudates [46]. However, it is difficult to evaluate the acute lesions of AD patients strictly by histopathological methods in clinical practice, especially for adult AD patients, as clinical symptoms in these patients are characterized by a chronic and recurrent course. Therefore, APT-positive reaction sites have been used for pathological analysis as a prototype of acute AD lesions exhibiting spongiotic dermatitis that reflects IgE-mediated delayed-type hypersensitivity [77,80]. Summarizing reports to date, the following characteristics have been noted.

In APT-positive reaction sites for HDM antigens in the non-lesioned skin of IgE-allergic AD patients (Figure 2c), HDM antigens are captured by IgE+ DCs (Figure 3e,f) [78,86]. In addition, epidermal keratinocytes secrete TSLP and induce production of the Th2 cytokines CCL17/TARC by DCs [91]. In vitro experiments revealed that IgE+ LCs have a much greater ability to present HDM antigens to lymphocytes than IgE− LCs, and there is a correlation between the incidence rates of IgE-expression on LCs in the skin and positive APT results [71]. Hence, IgE+ LCs are thought to initiate the formation of spongiotic dermatitis in APTs. The IgE+ LCs that capture HDM antigens migrate to the dermis over time, and 48 h after application, infiltration of lymphocytes and macrophages is observed, accompanied by basophils and eosinophils in the upper dermis and spongiosis formation in the thickening epidermis [78,92]. HDM antigens that have penetrated into the dermis are also captured by MCs, eosinophils, and macrophages [78], which induce the activation of infiltrating cells. In addition, macrophages that capture HDM antigens are thought to be a source of the secreted IL-12 that induces a shift to Th1 immunity [44]. In contrast, IgE+ IDECs, which are not seen in non-lesioned skin, become apparent in the upper dermis and infiltrate into the epidermis where they aggregate inside spongiosis areas and are finally eliminated outside the epidermis (Figure 3e,f) [77,80,86]. Lymphocytes infiltrating into areas of spongiosis consist primarily of CD4+ T cells, and some of these cells are CD4+ CD25+ Forkhead-box protein-P3 (FOXP3)+ T cells with regulatory functions [77,93]. In the analysis of allergen-specific T cell clones from the skin lesions of APT-positive reaction sites for HDM antigens, it was confirmed that both Th2 clones (secretion of IL-4, IL-13, etc.) and Th1 clones (secretion of IFN-γ) specific to HDM antigens could be established, with the former having the ability to produce IgE [79,81,94]. The Th1 cells and their secreted cytokine IFN-γ may activate the Fas and Fas-ligand interaction (Fas/FasL) system and induce keratinocyte apoptosis in the spongiosis area [95]. In addition, cell-mediated biphasic immune responses occur in HDM-induced APT-positive skin. A previous report showed that 24 h after induction of a positive reaction, expression of IL-4 mRNA, but not IFN-γ mRNA, was increased; 48–72 h after induction, almost no IL-4 mRNA was detected, but IFN-γ mRNA was significantly overexpressed in APT-positive skin [96]. A recent molecular profile analysis showed significantly increased expression of Th2, Th9, and Th17/Th 22 cytokines (i.e., IL-5, IL-13, IL-9, IL-17) and the Th1 chemokine CXCL10 in HDM-induced APT-positive skin (at 72 h after exposure) [97]. However, in this study, no significant increase in the expression of Th1 cytokines (i.e., IFN-γ) was observed.

Regarding the relationship between spongiosis and the localization of HDM antigens in APT-positive reaction sites, we reported a case in which HDM antigens were not captured by IgE+ IDECs in the central area of a spongiotic vesicle, but instead by IgE+ LCs in the area around the spongiotic vesicle (Figure 3f) [86]. However, when the same case was reexamined, it was confirmed that both IgE+ LCs and IgE+ IDECs that captured HDM antigens gathered in an early lesion of the spongiosis (Figure 3e) (unpublished data). These findings suggest that HDM antigens not only cause inflammation as an allergen, but they might be also rapidly treated and eliminated as foreign bodies associated with spongiosis formation.

##### Spongiotic Dermatitis in Lichenified Eczema

The results of our histopathological analyses suggested that a majority of lichenified eczema in AD may be chronic active lesions with areas of spongiotic dermatitis [76] (Figure 2d). This means that spongiotic dermatitis in lichenified eczema can be regarded as a sign of a newly occurring acute lesion within a chronic lesion. The following characteristics and pathogenic mechanisms emerge when these data are analyzed together with data from previous research. As mentioned above, analysis of infiltrating cells by H and E staining or ordinary immunostaining indicates that there is no significant difference in the tissue response of spongiotic dermatitis between AD and other types of eczematous dermatitis, such as ACD [75]. Although CD8+ T cells are considered to be more important in ACD [74], the cytokine profile induced by the pathomechanism of spongiosis is very similar across types of eczematous dermatitis. For the induction of apoptosis of keratinocytes, which is the key reaction in the development of spongiosis, the Fas/FasL system and cell-mediated cytotoxicity play important roles. The Fas/FasL system is induced by interactions among cytokines and molecules such as INF-γ and Fas ligand secreted by activated CD4+ T cells, Fas receptor (CD95) expressed on keratinocytes, and tumor necrosis factor (TNF)-α secreted by both the activated CD4+ T cells and keratinocytes, and cell-mediated cytotoxicity is induced by perforin and granzyme B, which are released primarily by CD8+ T cells [95,98,99]. However, in IgE-allergic AD, IgE-mediated delayed-type hypersensitivity could play a pivotal role in the development of spongiotic dermatitis. This is because in the spongiosis of chronic active lesions in IgE-allergic AD patients, aggregates of IgE+ IDECs are observed as a characteristic finding (Figure 3g,h), as in the case of spongiosis in APT-positive reaction sites. Furthermore, in our histopathological analysis, approximately 60% of the examined cases of IgE-allergic AD with HDM allergy exhibited accumulation of IgE+ IDECs that captured HDM antigens in the central areas of the spongiosis lesions (approximately 8 on average per 0.24 mm^2^ in a 200× visual field) (Figure 3i) [76]. These IgE+ IDECs were accompanied by infiltrating T lymphocytes, with a predominance of CD4+ cells (Figure 3j), and IgE+ LCs that captured HDM antigens were also found in the peripheral areas of the spongiosis lesions. Furthermore, in these cases, significantly greater numbers of IgE+ infiltrating cells that captured HDM antigens were also found in the upper dermis beneath the spongiotic epidermis. Similar findings were not observed in control subjects with other inflammatory skin diseases involving serum hyper-IgE. In addition, in control subjects with other types of eczematous dermatitis, IDECs not expressing IgE were also found in spongiotic dermatitis lesions; however, few HDM antigens were observed with these IDECs in the epidermis or with infiltrating cells in the upper dermis [76]. In contrast, previous studies have demonstrated that allergen-specific CD4+ and CD8+ T cell clones can be established from skin samples of chronic AD lesions in AD patients with HDM allergies, and a majority (71%) of these clones can produce INF-γ [100]. Therefore, these data strongly suggest that unlike spongiotic dermatitis caused by other diseases, the spongiotic dermatitis observed in lichenified eczema of patients with IgE-allergic AD is a pathological change based upon IgE-mediated delayed-type hypersensitivity.

Interestingly, we encountered a case of IgE-allergic AD with lichenified eczema, in which co-expression of HDM antigens was observed only with IgE+ LCs in the areas around spongiotic vesicles and not with IgE+ IDECs in the central area of the lesion (Figure 3k,l), similar to the preceding case of a patient with a spongiotic vesicle in the APT-positive reaction site [86]. In the preceding case of a positive APT result, it was also observed that IgE+ IDECs agglomerated in the spongiotic epidermis, but were expelled from the upper layer of the spongiotic vesicle to the outside of the skin after the APT (at 48 h after the test) for HDM antigens [86]. On the other hand, a previous report indicated that Langerhans cell microgranulomas (which can be considered aggregates of IDECs according to current data) in spongiotic dermatitis are removed by trans-epidermal elimination [82]. The remains of IgE+ IDEC agglomerates are often found in the scaly crust of the superficial layer of the spongiotic epidermis in AD lesions [86]. These findings suggest “a time course kinetics of spongiosis formation”, as follows: When the spongiosis formation that induced by the crosstalk among HDMs-antigens-capturing IgE+ LCs and IgE+ IDECs, infiltrating T lymphocytes, and keratinocytes have progressed sufficiently over time, and becomes a state of spongiotic vesicles with an epidermal dendritic cell aggregates, the detection sensitivity of HDMs-antigens coexisted with the IgE+ IDECs inside the spongiosis is reduced, and those IgE+ IDECs are removed by trans-epidermal elimination, while IgE+ LCs with HDMs-antigens migrate to the dermis or remain around the spongiosis for a while. Thus, if we consider spongiotic dermatitis as a type of biological defense response, then IgE+ IDECs could play a role as inflammatory DCs, both in the presentation of HDM antigens to lymphocytes as allergens and in treating, and rapidly eliminating, these antigens as foreign substances [101]. However, the pathogenetic mechanism of spongiotic dermatitis in IgE-allergic AD may not yet constitute a well-verified hypothesis. It is not so easy to establish a definitive and detailed relationship between spongiotic dermatitis and IgE-allergic AD, since factors such as those described above, i.e., factors other than IgE-mediated delayed-type hypersensitivity, may be involved in the manifestations of AD.

#### 4.2.2. Nature of, and Relationship between, Different Skin Manifestations of IgE-Allergic AD

By using schematics (Figure 4a,b), this section discusses the characteristics and relationship between non-lesioned and lesioned skin of patients with IgE-allergic AD. As mentioned above, in non-lesioned skin, IgE+ LCs are regularly arranged in preparation for allergen/antigen invasion in the epidermis, and an increase in the number of T cells expressing IL-13, IL-4, and INF-γ is observed in the dermis (Figure 4a: left-side part). In addition, IgE expression on MCs in the dermis is also increased [86,88]. These findings may be related to systemic molecular immune activation originating from the cytokine milieu of lesioned skin that feeds back to non-lesioned skin [45,79] (Figure 4b). Furthermore, keratinocytes in non-lesioned skin of AD patients have been reported to be more likely to undergo INF-γ–induced apoptosis than keratinocytes in normal skin of control subjects; in addition, *IFITMI* gene expression is significantly higher in keratinocytes of AD patients [102]. These characteristics of non-lesioned skin in patients with IgE-allergic AD clearly explain why these patients are more likely to experience activation of a specific immune response following allergen exposure, leading to eczematous dermatitis, and this could be the reason for the high rate of APT positivity in these patients (Figure 4a: right-side part). If non-lesioned skin is percutaneously stimulated with specific allergens, the lesions may transition to acute eczematous lesions [59]. However, it is presumed that this is less likely to occur because non-lesioned skin has a relatively well-preserved epidermal barrier function compared to lesioned skin [30,50]. Associated with increased inflammation in lesioned skin, symptoms of urticaria, urticarial erythema, or exudative inflammatory erythema might be more likely to occur in non-lesioned skin [25] due to the following: allergen invasion from dermal blood vessels following trans-gastrointestinal exposure to HDM allergens, as seen in pancake syndrome [103]; transairway exposure to HDM allergens, as seen in AD with asthma [104]; or trans-gastrointestinal/airway exposure to other allergens [1,47,60]. In general, patients with IgE-allergic AD exhibit polysensitization to environmental allergens, including airborne and food-derived allergens [60]. 

In lesioned skin (Figure 4b), lichenified eczema develops as a chronic inflammatory skin lesion caused by the interaction between cytokine/chemokine activation induced by innate immunity and allergic inflammation [47]. In this state, IgE+ LCs are present in the middle to upper layers of the epidermis, and IgE+ IDECs are present in the subepidermal layer of the epidermis in order to capture allergens that penetrate skin with an impaired barrier function, and IgE+ LCs primarily react to these allergens (Figure 4b: outer parts). In addition to the state of chronic inflammatory skin lesion, introduction of various inducing factors, such as increased specific-allergen exposure, innate immune danger signals (e.g., danger-associated molecular patters and pathogen-associated molecular patterns) resulting from skin barrier disruption and/or pathogen provocation, and release of proinflammatory cytokines (IL-1α, IL-1β, IL-18, and TNF-α) [7,47,105,106], can result in spongiosis dermatitis as an acute lesion in lichenified eczema. This is caused by the aforementioned pathomechanism, i.e., induction of keratinocyte apoptosis by the interaction between activated IgE+ LCs and accumulated IgE+ IDECs that captured allergens such as HDMs, the Fas/FasL system stimulated by INF-γ secreted by activated CD4+ T cells, and perforin and granzyme B released by activated CD4+ and CD8+ T cells (Figure 4b: central part). The accumulation of IgE+ IDECs in the central spongiosis lesion may be the result of mobilization of these cells from the lower epidermis and upper dermis. For the convergence of inflammation, CD4+ CD25+ FOXP3+ T cells with regulatory function might infiltrate the spongiotic epidermis as in a positive APT [77,93]. In addition, it has been confirmed that cleavage of adhesion molecules (i.e., E-cadherins) occurs in spongiosis associated with keratinocyte apoptosis [107]. It has also been reported that in the Fas/FasL system induced by INF-γ stimulation, a Fas receptor (CD95) density of 40,000 per keratinocyte is required as the threshold for keratinocyte apoptosis; these keratinocytes release Th1 chemokines such as IP-10 (CXCL10), Mig (CXCL9), and iTac (CXCL11) to attract Th1 cells that express the specific receptor CXCR3 [95,108]. It is also known that INF-γ stimulation enhances the expression of major histocompatibility complex (MHC)-I and MHC-II molecules in keratinocytes [47].

In general, infiltration of Th1 cells requires IL-12 secretion within lesion tissues, and macrophages, eosinophils, and DCs (especially IDECs) are known sources of this cytokine [45,109]. However, it has not yet been confirmed by in vivo analyses whether IDECs that capture allergens actually release IL-12 within the spongiotic epidermis. Alternatively, a recent study using RNA in situ hybridization reported that 5.75% (*n* = 7, range; 1.8 to 13.7%) of T cells infiltrating into the spongiotic epidermis express IL-13 [110]. INF-γ stimulation can also induce keratinocytes to produce the Th2 chemokine CCL22 [47]. Hence, these data suggest that both Th1 and Th2 cells might be involved in the development of spongiosis lesions.

Antigens in allergic skin diseases such as ACD are generally presented by MHC-class II (human leukocyte antigen–DR) on keratinocytes as haptens with carrier proteins, and lymphocytes react with these antigens to form spongiosis lesions [74]. Therefore, in IgE-allergic AD, whether allergens such as HDMs can be presented by keratinocytes, and if so, how the antigens are presented by these cells, remain interesting questions. There are two possibilities in this regard, one is that HDM antigens can also function as haptens [111], and the other is that the protein antigens of HDMs can be presented by MHC-class I and MHC-class II molecules on keratinocytes [112].

### 4.3. Pathomechanisms of Non-IgE Allergic AD and Indeterminate Allergic AD

In general, AD is roughly divided into two distinct variants depending on the presence or absence of involvement in the pathology of IgE allergy: (1) extrinsic IgE-allergic AD, which occurs in the context of sensitization to environmental allergens that produce allergen-specific IgEs and is accompanied by elevated serum total IgE levels (more than approximately 150 or 400 IU/mL, according to standards of the individual facility), and (2) intrinsic non–IgE-allergic AD, with no detectable sensitization to environmental allergens and with normal serum total IgE levels [113]. In the majority of patients with infantile and pediatric AD, the transition from non–IgE-allergic AD in the early stage to IgE-allergic AD in the advanced stage is a common disease course [41]. However, there are many cases of non–IgE-allergic AD in which this transition is not seen in female cases of adolescent and adult AD [21]. Numerous recent research papers have classified these two subtypes only by the level of serum total IgE (e.g., a cut-off value of 150 IU/mL); however, there is a concern that indeterminate allergic AD, which is a vague disease state and intermediate type of the above two variants, may be incorporated into non–IgE-allergic AD by this classification method. When considering the pathomechanisms of AD by IgE-associated immunopathology, indeterminate allergic AD with a positivity for allergen-specific IgEs and normal levels of serum total IgE should not be ignored as a third variant [25,33].

The concept of spongiotic dermatitis associated with IgE-mediated delayed-type hypersensitivity described above is based upon the presence of IgE+ LCs and IgE+ IDECs [55,56]. Previous studies have shown that IgE expression on DCs in the epidermis may occur when a patient’s serum total IgE level is at least 300 IU/mL (kU/L) [87] and definitely occurs when a patient’s serum total IgE level is >4000 IU/mL [71]. Therefore, it is considered that in patients with non–IgE-allergic AD and indeterminate allergic AD, spongiotic dermatitis would develop independently of IgE-mediated delayed-type hypersensitivity. However, some patients with indeterminate allergic AD have specific IgEs against pathogen-related antigens, such as enterotoxins of *S. aureus*, as well as environmental allergens, such as HDMs, pollen, and fungi [47,114,115]. Therefore, in indeterminate allergic AD patients, it is speculated that specific IgEs against allergens/antigens that can act as triggers for type I immediate responses and late-phase reactions might induce development of the symptoms, similar to protein contact urticaria and protein contact dermatitis [67,68]. These symptoms might modify the existing chronic AD symptoms caused by dysregulation of innate immunity and/or allergic inflammation, including the standard (IgE-independent) delayed-type hypersensitivity [60].

With regard to the characteristics of patients with non–IgE-allergic AD, the following cytokine profiles of serum and skin lesions have been reported. In comparison to patients with IgE-allergic AD, serum IL-13 levels are higher in patients with non–IgE-allergic AD, whereas serum IL-5 levels are higher in patients with IgE-allergic AD, and the serum IL-4 and soluble IL-4Rα levels are the same in these patients [43]. In contrast, studies indicate that inflammatory infiltrating cells expressing IL-5 and IL-13 in skin lesions and the capacity of lesional T cells to produce these cytokines are reduced in patients with non–IgE-allergic AD [38]. Based on PT and APT results in patients with non–IgE-allergic AD, it has been reported that metals and HDMs can be positive in the tests [116], and intraepidermal infiltration of IDECs that do not express IgE has also been confirmed in APT-positive reaction sites for HDM antigens [80]. Therefore, it is speculated that the spongiotic dermatitis observed in eczematous dermatitis of patients with non–IgE-allergic AD might be caused by haptens and allergens (e.g., metals and HDMs) that induce standard (IgE-independent) delayed-type hypersensitivity or an immune response similar to systemic-type contact dermatitis [60]. Alternatively, an immune response similar to irritant contact dermatitis in which innate immunity is activated by environmental irritants (e.g., chemicals) might develop in the spongiotic dermatitis lesions of these patients [117]. These are areas in which future research and analysis are expected.

## 5. Conclusions

It is expected that clinical practice and medical treatment of AD will be carried out in line with the understanding and stratification of endotypes and phenotypes based on molecular taxonomy data [1,17,60,118,119]. As mentioned above, AD is a complex pathogenetic disease in which a cytokine milieu dominated by type-2 (IL-4/IL-13, etc.) immunity and antigen-specific allergic inflammation are linked by various factors. Understanding the pathomechanism of eczematous dermatitis in AD as an allergic disease from the perspective of molecular science would be also important for realizing preventive and personalized therapies in the future [120]. Sulzberger, a pioneer of the atopic concept in AD, hypothesized that an unknown “factor X” bridges the gap between the IgE allergy (type I reaction) and delayed-type hypersensitivity (type IV reaction) of eczematous dermatitis in AD [53]. Current understanding suggests that this factor is IgE-mediated delayed-type hypersensitivity. However, the research findings presented in this review need further verification in future studies.

## Figures and Tables

**Figure 1 ijms-23-06682-f001:**
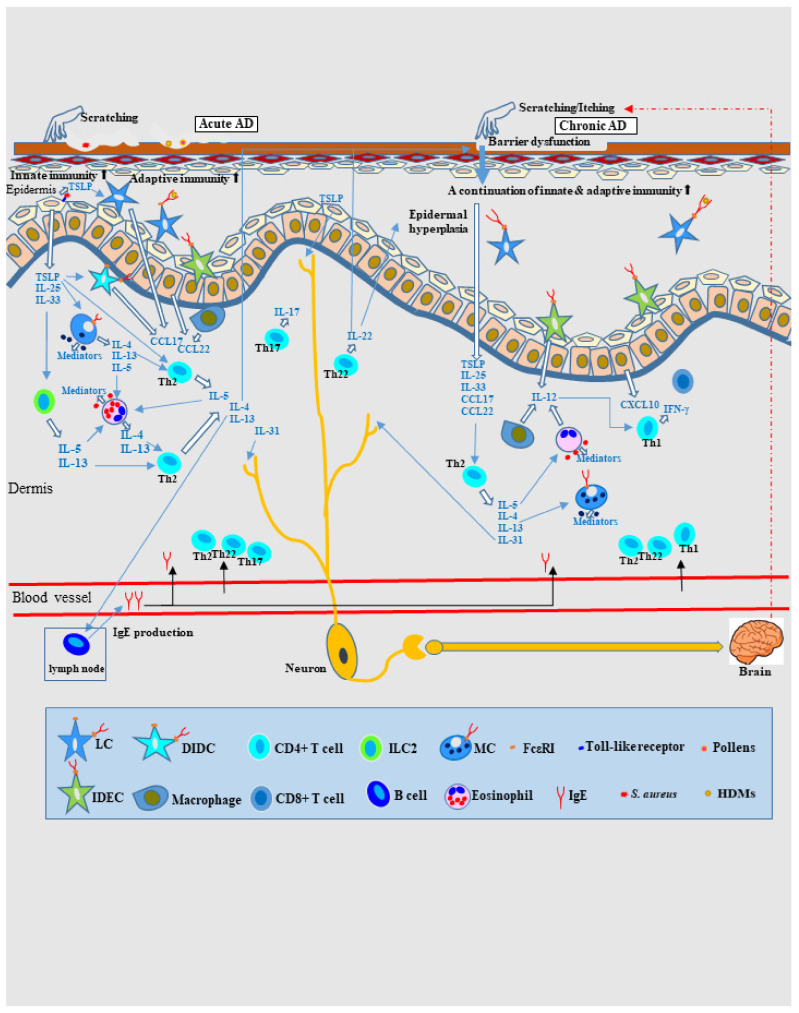
**Schematic illustration of the cytokine milieu of type-2**–**dominant immunity axes in AD.** In AD, both innate and adaptive immune responses are activated. Environmental stimuli (e.g., scratching and dysbiosis involving *S. aureus*) induce the release of type-2 (Th2) inflammation-promoting chemokines (e.g., TSLP, IL-25, and IL-33) from keratinocytes, and innate effector cells (e.g., DCs, MCs, and ILC2s) are activated to secrete type-2 cytokines and chemokines, ultimately inducing the recruitment of Th2 cells in lesioned skin. In adaptive immunity, T cell-mediated biphasic immune reactions develop. Th2 cytokines (i.e., IL-4, IL-5, IL-13, and IL-31), Th22 cytokines (i.e., IL-22), and Th17 cytokines (i.e., IL-17) are involved in acute lesions, and progressive activation of Th2 and Th22 cytokines and significant upregulation of Th1 cytokines (i.e., IFN-γ) occur in chronic lesions. For the recruitment of Th1 cells, IL-12 secretion is needed in lesioned skin, and macrophages, eosinophils, and DCs are thought to be sources of these cytokines. In both acute and chronic lesions, Th2 cytokines play many roles: for example, IL-4 and IL-13 stimulate B cells to produce IgE and promote IgE binding to MCs and DCs (e.g., LCs and IDECs); IL-5 increases eosinophil viability; and IL-31 and TSLP cause itching by directly acting on peripheral nerves. In addition, IL-4, IL-13, and the Th22 cytokines IL-22 induce epidermal barrier dysfunction. The Th22 cytokine IL-22 also induces epidermal hyperplasia in chronic lesions. Abbreviations: AD, atopic dermatitis; CCL, CC chemokine ligand; CD, cluster of differentiation; CXCL, CXC chemokine ligand; DC, dendritic cell; DIDC, dermal inflammatory dendritic cell; FcεRI, high-affinity IgE receptor; HDMs, house dust mites; IDEC, inflammatory dendritic epidermal cell; IgE, immunoglobulin E; IL, interleukin; ILC2, type-2 innate lymphoid cell; LC, Langerhans cell; MC, mast cell; *S. aureus*, *Staphylococcus aureus*; Th, T helper; and TSLP, thymic stromal lymphopoietin.

**Figure 2 ijms-23-06682-f002:**
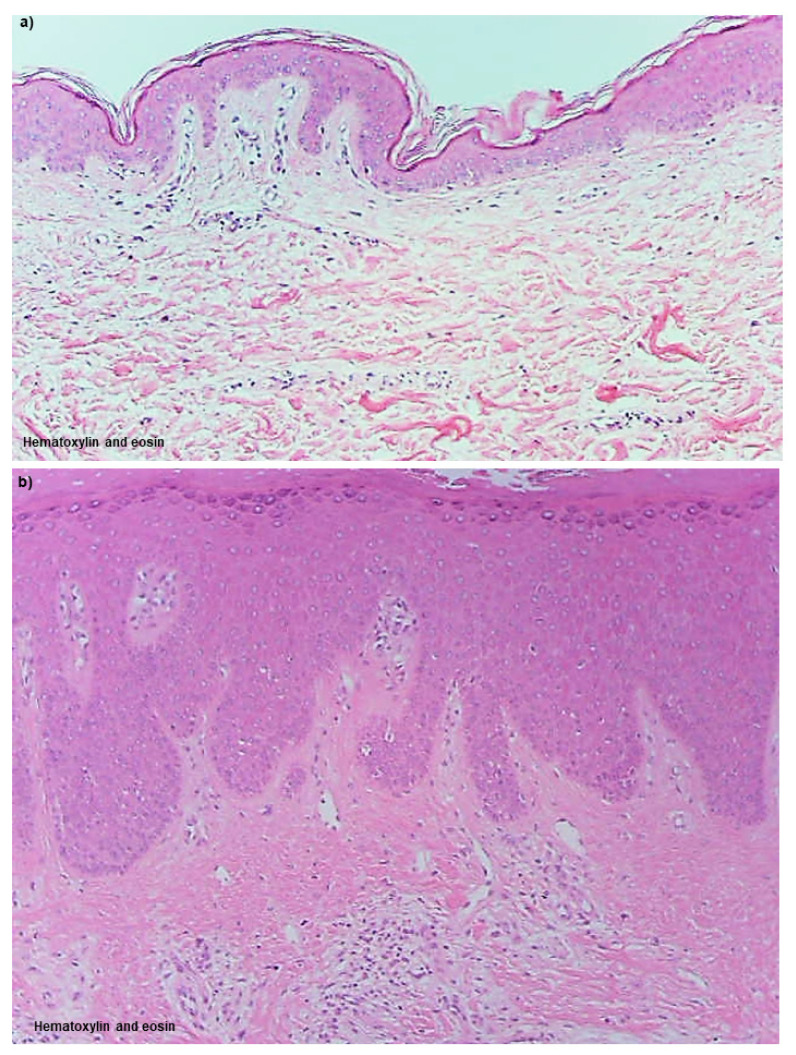
**Histopathology of skin manifestations of IgE-allergic AD.** (**a**). **Uninvolved skin**: No abnormal findings are observed by hematoxylin and eosin staining. (**b**). **Lichenified eczema; non-spongiotic dermatitis**: Epidermal hyperplasia with an irregular extension of the rete ridges, and dermal infiltration of lymphocytes and macrophages with increased MCs are observed. (**c**). **Atopy patch test; spongiotic dermatitis**: Focal spongiosis with infiltration of small round cells is observed in the middle to lower layers of the epidermis. (**d**). **Lichenified eczema; spongiotic dermatitis**: Focal spongiosis with infiltration of small round cells is observed in the acanthotic epidermis. Original magnifications: 100×, (**a**–**d**). Characteristics of the patients shown the figures: the patient was a 61-year-old woman with an elevated serum total IgE level of 10,198 IU/mL and specific IgEs for Der f and Der p, (**a**,**b**,**d**); and the patient was an 84-year-old man with an elevated serum total IgE level of 19,757 IU/mL and specific IgEs for Der f and Der p (**c**). Abbreviations: AD, atopic dermatitis; Der f, *Dermatophagoides farinae*; Der p, *Dermatophagoides pteronyssinus*; IgE, immunoglobulin E; and MCs, mast cells.

**Figure 3 ijms-23-06682-f003:**
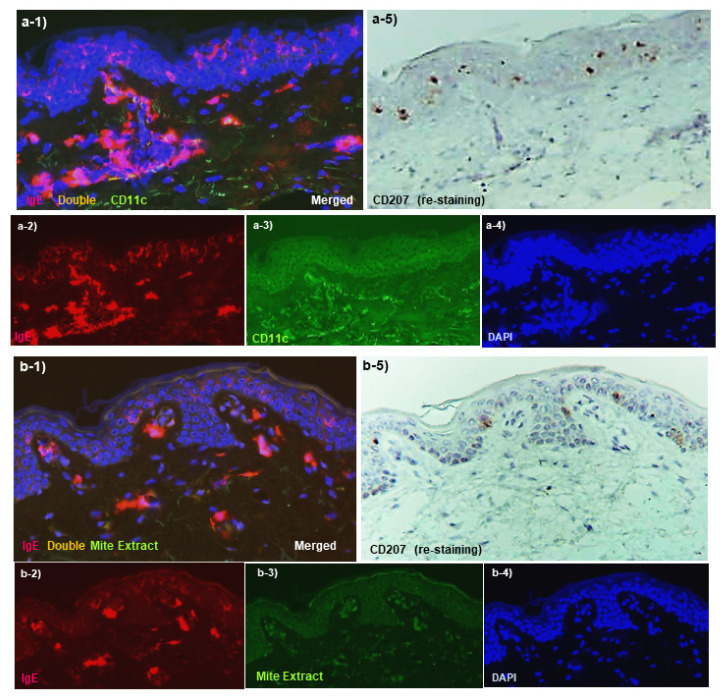
**Immunohistopathology of skin manifestations of IgE-allergic AD.** (**a**). **Uninvolved skin:** Regularly arranged IgE+ CD207+ LCs (red images of **a-1** that colocalize with brown images of **a-5**), but not infiltrated IgE+ CD11+ IDECs (yellow image of **a-1**), are observed in the epidermis. Infiltration of IgE+ CD11+ inflammatory DCs (yellow images of **a-1**) in the upper dermis is slight. Note that the dendrites of LCs are not stained with CD207 (**a-1**,**a-5**).(**b**). **Uninvolved skin:** Few HDM antigens captured by IgE + LCs in the epidermis and by IgE+ infiltrating cells in the upper dermis are observed. (**c**). **Lichenified eczema; non-spongiotic dermatitis:** In the epidermis, IgE+ CD207+ LCs (red images of **c-1** that are colocalized with the brown images of **c-5**) are located in the upper-middle layer, and IgE+ CD11+ IDECs (yellow images of **c-1**) are located in the lower layer. LCs without IgE expression (brown images of **c-5** that are not colocalized with the red images of **c-1**) are also observed in the epidermis. (**d**). **Lichenified eczema; non-spongiotic dermatitis:** Most of the HDM antigens found in the epidermis are captured by IgE+ LCs (yellow images of **d-1** that are colocalized with the brown images of **d-5**, arrows). An IgE+ LC that captured HDM antigens by extending a dendrite to the tight junction side is also observed (thick arrows in **d-1** and **d-5**). Minimal lymphocyte infiltration is observed around these HDM antigen-capturing LCs. HDM antigens are also captured by IgE-expressing infiltrating cells, which are different from LCs, in the papillary and upper layers of the dermis (yellow images of **d-1** that are not colocalized with the brown images of **d-5** in the papillary and upper dermis). (**e**). **Atopy patch test; an early lesion of focal spongiosis:** Focal spongiosis shows an agglomerate infiltrated by IgE+ CD207+ LCs (arrows in **e-1** and **e-5**) and IgE+ CD207− DCs (IDECs) (arrowheads in **e-1** and **e-5**). In this agglomerate, both IgE+ LCs that captured HDM antigens (Der f1) (yellow images of **e-1** that are colocalized with the brown images of **e-5**, yellow arrows) and IgE+ IDECs that captured HDM antigens (yellow images of **e-1** that are not colocalized with the brown images of **e-5**, yellow arrowheads) are observed. (**f**). **Atopy patch test****; spongiotic vesicle:** Inside the spongiotic vesicle, IgE+ CD207− DCs (IDECs) (red images of **f-1** that are not colocalized with the brown images of **f-5**, white arrowheads) form an agglomerate, but HDM antigens (Der f1) captured by these cells are rarely observed. In the peripheral areas of the spongiotic vesicle, IgE+ LCs that captured HDM antigens (yellow images of **f-1** that are colocalized with the brown images of **f-5**, yellow arrows) are observed. (**g–j**). **Lichenified eczema; spongiotic dermatitis:** IgE+ CD11+ IDECs infiltrate and aggregate in focal spongiosis (yellow images in the epidermis of **g-1**). IgE+ CD206+ IDECs infiltrate and aggregate in focal spongiosis (yellow images in the epidermis of **h-1**). Most of the IgE+ CD206+ IDECs that infiltrate in focal spongiosis capture HDM antigens (Mite extract) (yellow images in the epidermis of **i-1**). Accompanying infiltration of CD3+ CD8− (CD4+) T cells is mainly seen in this focal spongiosis (green images in the epidermis of **j-1**). (**k,l**). **Lichenified eczema; spongiotic dermatitis that forms a spongiotic vesicle:** Inside the spongiotic vesicle, IgE+ CD1a+ CD207− DCs (IDECs) form an agglomerate (red images in the center of the epidermis in **k-1**,**l-1**), but HDM antigens (Der f1) captured by these cells are rarely observed. IgE+ CD1a+ CD207+ LCs (arrows in **k-1**,**k-5**,**l-1**,**l-5**) are found around the spongiotic vesicle, and some of these cells captured HDM antigens (Der f1) (yellow arrows in **k-1**,**l-1**). Original magnifications: 200×, (**a-l**). In double immunofluorescence staining, nuclei are labeled with 4′,6-diamidino-2-phenylindole (DAPI, blue images). The images (**f**–**i**,**k**,**l**) are reproduced with permission from the authors and journals (*Dermatopathology* and *Dermatology Clinics and Research*) [76,86]. Characteristics of the patients shown the figures: the 61-year-old woman, (**a**–**d**,**g**–**j**); and the 84-year-old man, (**e**,**f**) were the same patients described in Figure 2; and the patient was a 71-year-old man with an elevated serum total IgE level of 2413 IU/mL and specific IgEs for Der f (**k**,**l**). Atopy patch test was carried out using allergen extracts of Der f [86]. The primary polyclonal antibodies against HDM antigens were mite extract (mite crude extract containing antigens from Der f and Der p) and Der f1 (the primary allergenic component of Der f that cross-reacts with Der p1) [76]. Abbreviations: AD, atopic dermatitis; CD, cluster of differentiation; Der f, *Dermatophagoides*
*farinae*; Der p, *Dermatophagoides pteronyssinus*; DC, dendritic cell; HDMs, house dust mites; IDEC, inflammatory dendritic epidermal cell; IgE, immunoglobulin E; LC, and Langerhans cell.

**Figure 4 ijms-23-06682-f004:**
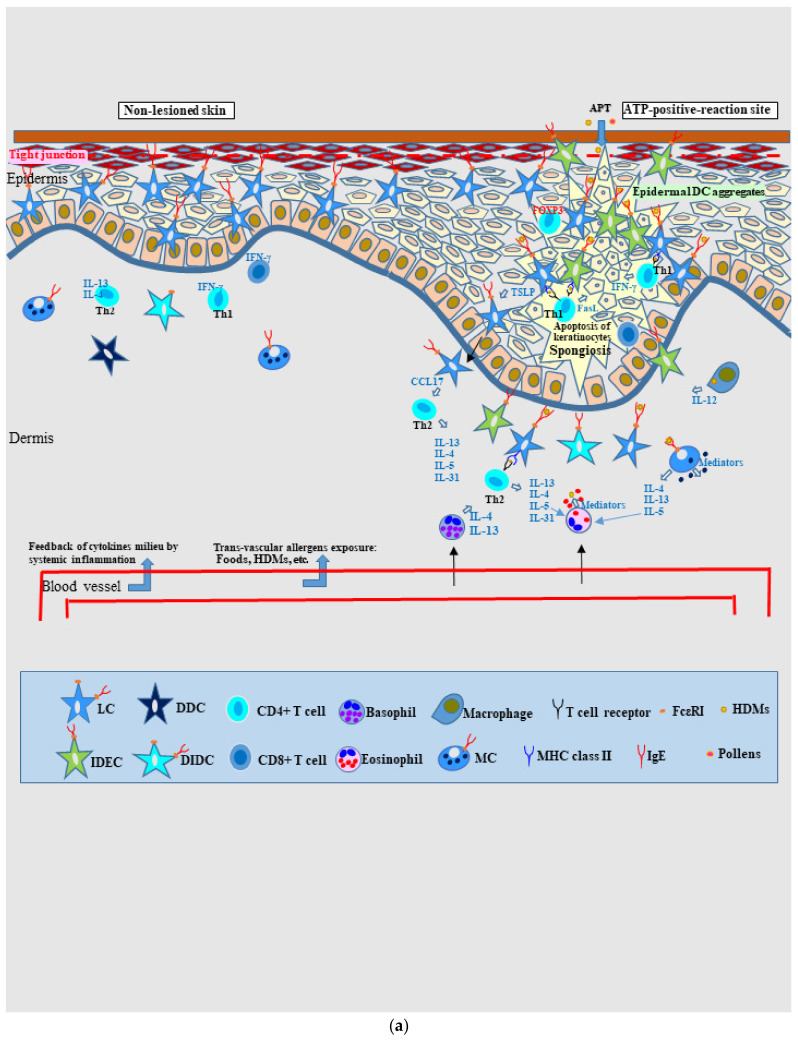
(**a**). **Schematic illustration of non-lesioned skin and the positive-reaction site of atopy patch test in IgE-allergic AD.** In non-lesioned skin, IgE-expressing LCs are regularly arranged in preparation for allergen/antigen invasion of the epidermis, and both Th2 (IL-4– or IL-13-expressing) and Th1 (INF-γ-expressing) cells and T cytotoxic (INF-γ-expressing CD8+) cells are increased among dermal infiltrating cells. IDECs are not found in the epidermis, and inflammatory DCs expressing IgE are almost undetectable in the dermis. Allergens such as HDMs are rarely observed in the dermis and epidermis by histopathological analysis. In the APT-positive reaction site for allergens (e.g., HDMs and pollens), spongiotic dermatitis is observed as a characteristic finding. Therefore, this characteristic has been used in pathological analysis as a prototype of acute AD lesions that reflect IgE-mediated delayed-type hypersensitivity. Considering the APT for HDM antigens in IgE-allergic AD, the following characteristics have been revealed: HDM antigens are initially captured primarily by IgE+ LCs; at 48 h after application, spongiosis formation in the epidermis and infiltration of lymphocytes and macrophages accompanied by basophils and eosinophils in the upper dermis are observed; keratinocytes secrete TSLP and induce the production of the Th2 cytokine CCL17 by DCs (probably LCs); penetrated HDM antigens are also captured by MCs, eosinophils, and macrophages, and the activated macrophages secrete IL-12, which induces the recruitment of Th1 cells; IgE+ IDECs become apparent in the upper dermis and aggregate inside the spongiosis lesion; in early spongiosis, both HDM antigen-capturing IgE+ IDECs and IgE+ LCs are seen in the epidermal DC aggregates; infiltrating lymphocytes are primarily CD4+ T cells, but some are CD4+ CD25+ FOXP3+ T cells with regulatory functions; and keratinocyte apoptosis possibly induced by IFN-γ-secreting Th1 cells and the Fas/FasL system is the key reaction in spongiosis formation. (**b**). **Schematic illustration of areas of spongiotic dermatitis and non-spongiotic dermatitis in lichenified eczema of IgE-allergic AD.** Lichenified eczema develops as a chronic inflammatory skin lesion caused by the interaction between cytokine/chemokine activation by innate immunity and allergic inflammation. In this state, to prepare for capturing allergens invading through the impaired epidermal barrier, IgE+ LCs are present in the middle to upper epidermis, and IgE+ IDECs are present in the lower epidermis, and IgE+ LCs primarily react to the allergens (non-spongiotic epidermis; both outer parts of the schematic). Following the introduction of several inducing factors, spongiosis dermatitis reflecting IgE-mediated delayed-type hypersensitivity may occur as an acute lesion in lichenified eczema (spongiotic epidermis; central part of the schematic illustration). The following pathomechanism is possible: IgE+ LCs and IgE+ IDECs that capture allergens (e.g., HDMs) become activated, and the IgE+ IDECs gather in the central area of the spongiosis lesion and form an epidermal DC aggregate; the IgE+ epidermal DCs (i.e., LCs and IDECs) induce the infiltration of allergen-specific CD4+ T cells in the spongiotic epidermis; the Fas/FasL system is induced by the interactions between the IgE+ epidermal DCs, activated CD4+ T cells, and keratinocytes to induce the expression of TNF-α, INF-γ, Fas ligand, and Fas receptor (CD95); cell-mediated cytotoxicity is also induced by perforin and granzyme B, which are primarily released by CD8+ T cells; keratinocyte apoptosis is the key reaction in the development of spongiosis lesions and progresses to completion under control of the Fas/FasL system, cell-mediated cytotoxicity, and T cells with regulatory function. Abbreviations: AD, atopic dermatitis; APT, atopy patch test; CD, cluster of differentiation; DC, dendritic cell; DDC, dermal dendritic cell; DIDC, dermal inflammatory dendritic cell; FasL, Fas-ligand; FcεRI, high-affinity IgE receptor; FOXP3, Forkhead box protein P3; HDMs, house dust mites; IDEC, inflammatory dendritic epidermal cell; IgE, immunoglobulin E; IL, interleukin; IFN, interferon; LC, Langerhans cell; MC, mast cell; MHC, major histocompatibility complex; *S. aureus*, *Staphylococcus aureus*; Th, T helper; TNF, tumor necrosis factor; and TSLP, thymic stromal lymphopoietin.

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
