# Peer review of "Immunological Pathomechanisms of Spongiotic Dermatitis in Skin Lesions of Atopic Dermatitis"

_ijms, 2022, doi:10.3390/ijms23126682_

Round 1
Reviewer 1 Report
ijms-1748994
Immunological pathomechanisms of spongiotic dermatitis in skin lesions of atopic dermatitis
This is a Review paper oriented to summarize the most relevant aspect of the complex related to atopic dermatitis.
Despite the essential lines conducted in this Review, some aspects should be considered:
1. In the Introduction section (lines 45 and 46) it is very much difficult to understand:
“…therefore, AD is considered an allergic disease that can become a lifelong condition.”
We have to recognize that we need still more evidence to confirm that atopic dermatitis can be considered as an allergic disease. Even more, some authors propose that atopic dermatitis is a previous step to allergic disease (food allergy, environmental…).
2. To propose that the pathogenic mechanism of spongiotic dermatitis is based on atopic dermatitis still constitutes a weak hypothesis.
3. It is difficult to establish a close relationship between spongiotic dermatitis “in adult” with atopic dermatitis, due to the multiple factors involved in the appearance of atopic eczema. Perhaps the “atopic eczema in the elderly” could be more extensively trated.

Reviewer 2 Report
This review has included a comprehesive overview for the pathomechanism of spongiotic dermatitis. It is very good organization except some format issues and the figures. Some of the English is akward.
The figures should insert to the manuscript as the other published papers in this journal. The author should re-arrange the figures in figure 2. Maybe a table could help somehow.
The reference format in the texts should be consitent, such as [6, 7] and [6-7].
Staphylococcus aureus should be Italic. The author have to read through the manuscript to revise other similar issues.
(121) should be [121].
Round 2
Reviewer 1 Report
The authors have followed the most indications suggested.